# COVID-19 Confinement Effects on Game Actions during Competition Restart in Professional Soccer Players

**DOI:** 10.3390/ijerph19074252

**Published:** 2022-04-02

**Authors:** Abraham García-Aliaga, Moisés Marquina, Ignacio Refoyo Román, Diego Muriarte Solana, Juan A. Piñero Madrona, Roberto López del Campo, Fabio Nevado Garrosa, Daniel Mon-López

**Affiliations:** 1Facultad de Ciencias de la Actividad Física y del Deporte (INEF-Sports Department), Universidad Politécnica de Madrid, 28040 Madrid, Spain; abraham.garciaa@upm.es (A.G.-A.); diego.muriarte@upm.es (D.M.S.); daniel.mon@upm.es (D.M.-L.); 2Facultad de Ciencias del Deporte, Universidad Castilla La-Mancha, 45004 Toledo, Spain; juanpf79@hotmail.com; 3Department of Competitions and Mediacoach, LaLiga, 28043 Madrid, Spain; rlopez@laliga.es (R.L.d.C.); fnevado@laliga.es (F.N.G.)

**Keywords:** lockdown, performance, playing position, physical demands, GPS, football

## Abstract

The main objective of the present study was to compare high-intensity actions in a week of three matches before and after the COVID-19 lockdown. The observational methodology was used. This study analysed 551 professional soccer players from 22 different Spanish teams (LaLiga Smartbank 2019–2020) by a multi-camera tracking system and associated software (Mediacoach^®^, Spain). Variables of distances per minute and totals, travelled at High Intensity (HIR), Very High Intensity (VHIR), Sprint (HSR), player’s maximum speed, average speed, and the number of efforts in VHIR and HSR were analysed in the first and second half of the games, the full match, as well as in relation to the playing position. Players who participated in the same number of matches pre- and post-COVID-19 showed an increase in the total minutes played, *p* < 0.05, and small decreases in game actions, *p* < 0.05, with an effect size between 0.21 and 0.45, while players who participated in different number of matches pre- and post-COVID-19 showed a performance decrease, *p* < 0.05, with a size effect between 0.13 and 0.51; this was evident, particularly, for midfielders, *p* < 0.05, with a size effect between 0.39 and 0.75. The results seem to show that the playing intensity after COVID-19 confinement did not lead to large performance losses, except for midfielders who were the most involved players and showed a higher decrease in performance. The main findings of this study could provide insight to football coaches for rotations in starting line-ups and game substitutions, so as not to affect the intensity levels of the competitions.

## 1. Introduction

The COVID-19 pandemic has caused a global emergency. Many countries have been forced to impose a state of quarantine or confinement in which freedom of movement was restricted. This health crisis has had an impact on the sports industry, specifically, in soccer leagues [1]. Accordingly, professional soccer players have not been able to train regularly, causing the postponement or suspension of competitions [2,3]. Thus, their training conditions and perceptions have been affected [4]. In addition, the training protocols changed during the confinement and the reincorporation in teams, with workouts carried pout in isolation, without opposition, and in very different situations with respect to competitions [5].

Because of the state of alarm decreed by the Spanish government and the subsequent lockdown of the population [6], professional footballers were able to return to their clubs’ installations only on 11 May 2020. From this date, the footballers were able to start their training, first using football-specific training routines while maintaining a “social” distance of 1 to 2 m (for 1 week) and then using reduced training groups. Therefore, professional football players competing in LaLiga™ were confined to their homes for 8 weeks and were allowed to train in preparation for the first match of the competition for 4 full weeks, after a total suspension of competitions that lasted 12 weeks. In addition, no friendly matches could be played. Considering these different ways of training, it seems appropriate to think that players’ post-lockdown performance might decrease.

Soccer is a complex sport of interaction between players (collaboration–opposition). Footballers used to make random transitions between maximum or submaximal efforts, high-intensity multidirectional displacements, and longer periods of low-intensity activity [7]. The normal distance covered per match ranges between 9 and 14 km, with approximately 10% of high-intensity running [8]. However, these data depend on each specific playing position [9,10] and on the intensity of the actions defined by the characteristics of each specific match.

With regard to it, intermittent actions can be divided into different intervals using speed thresholds: high-intensity runs (14–21 km/h), very high intensity runs (21.1–24 km/h), and sprints (more than 24 km/h). The distances run at higher speeds are progressively shorter as the speed ranges increase [11], being 1614 ± 320 m for high-intensity runs [12], 847 ± 349 m for very high intensity runs [13], and 184 ± 87 m for sprints [14]. Interestingly, footballers’ performance has been associated with the maximum speeds reached and the distance travelled at very high intensities, being the number of sprints, as well as the maximal speed during the match, crucial [15].

Previous literature has concluded that distances run at high intensity and sprints are the variables with the highest coefficients of variation during a season, while total distance and low-intensity activities show smaller variations [16]. Furthermore, it was found that variability was also different between playing positions, as central defenders showed the highest coefficients for high-intensity activities [9,16,17].

The lockdown caused by the COVID-19 pandemic represented a major factor inducing physiological changes in elite footballers [18]. The return to sport of professional footballers occurred after an enforced lockdown never experienced before and longer than the normal annual season break. Moreover, this lockdown could affect the performance and injury risk of the players [19,20]. It is important to mention that the competition schedules and the increased match congestion, with a high frequency of matches separated by less than 72 h, could influence the volume and intensity of the players during the matches. Thus, in short-term matches, low- and medium-intensities and sprint distances were significantly different, and these differences were also apparent depending on the playing position [21]. The influence of periods with congested matches has been proved especially in relation to the injury rate [14,22]. It was found that neither the total distance covered nor the distance covered at lower intensities had greater variability across matches, while high-intensity running and injury risk were unaffected by a prolonged period of match congestion [23]. In addition, match congestion after the first COVID-19-related lockdown did not have a significant impact on muscle injury rate in professional players [24]. However, some authors detected more injuries caused by COVID-19 lockdown during the 2020–2021 season than in the previous two seasons, which could indicate a long-term adverse effect, and may be related to match congestion [25]. Therefore, the influence of the COVID-19 lockdown is somehow controversial, as there is no evidence of a decrease in physical performance in the post-lockdown period compared to the pre-lockdown period. In addition, there is no proof either on whether a decrease in physical performance is caused by the accumulation of games (congested time) [26].

Taking everything previously mentioned into account, it seems advisable to compare the load and intensity values of the players in relation to their specific positions in a week of three games before and after the confinement. Therefore, the purpose of this study was to determine whether the intensity of the actions was reduced in a week with three matches due to the modification in the soccer players’ training conditions caused by the confinement.

## 2. Materials and Methods

### 2.1. Participants

The participants in the study were a total of 511 players from 22 teams. Only professional players belonging to the Second Spanish Soccer Division (LaLiga Smartbank 2019–2020) were analysed. The classification by game positions was carried out following the recommendations by Di Salvo et al. [10] (see Figure 1). The players in our study were distributed as follows: 93 central defenders (CD), 83 external defenders (ED), 134 central midfielders (CM), 110 external midfielders/wingers (W), and 91 forwards (F).

As exclusion criteria, only players who completed the full game time (90 min + discount) were considered [27,28]. In addition, goalkeepers were excluded, since their physical demands cannot be compared with those of the rest of the players. After applying the exclusion criteria, the final descriptive sample distribution is shown in Table 1.

### 2.2. Methodology

The present study is observational research. The data were obtained from the Official Tracking System of the Spanish league “LaLiga Smartbank”. LaLiga^TM^ authorised the use of the variables included in this research. The data were retrieved by using a valid and reliable multi-camera tracking system and associated software (Mediacoach^®^, Madrid, Spain) [29]. The Mediacoach^®^ physical performance data records the position of each player using a stereo multi-camera system consisting of two multi-camera units located on both sides of the centre line of the field (TRACAB^®^, Stockholm, Sweden) that has a sampling frequency of 25 Hz. Each multi-camera unit contains three cameras that are synchronised in order to provide a stitched panoramic image with a resolution of 1920 × 1080 pixels [30]. This tracking system semi-automatically assessed the match performance data of all players, the position of the ball, and the corresponding match events, allowing the intensity of the athlete’s run to be shown. The validity of Mediacoach^®^ to assess the running distance during a soccer match play was obtained along with data obtained with Global Positioning System units [31] and with further data obtained from a reference camera system (i.e., VICON motion capture system) [32].

Rounds of league numbers 8th, 9th and 10th (pre-COVID-19 isolation period) and numbers 32nd, 33rd and 34th (post COVID-19 isolation period) were analysed. These rounds of the league were selected by applying the following criteria: (A) rounds 8th, 9th and 10th were the only ones played in a window of 3 games in the same week before the isolation period, (B) rounds 32nd, 33rd and 34th were the first three post COVID-19 rounds with the same time matches structure as the 8th, 9th and 10th.

The analysis of the periods was carried out by structuring the matches as follows: (1) group of balanced matches (same number of matches before and after confinement). This group included the players who participated before and after the isolation period in one, two or three matches, (2) group of unbalanced matches (different number of matches before and after confinement). This group included the players who participated in matches before and after the isolation period, but in a different number of matches in each period. This study was approved by the Ethics Committee of the Polytechnic University of Madrid. The authors have no conflicts of interest to report.

### 2.3. Variables

Variables that inform about the physical response in competition were considered as dependent variables following previous classifications [28,30,33]: minutes played, total distance travelled (DT), total distance travelled per minute (relative distance m/min), distance travelled at high intensity (HIR) (14–21 km/h), distance travelled at very high intensity (VHIR) (21–24 km/h), distance travelled at sprint (HSR) (>24 km/h), player’s maximum speed, average speed, and number of efforts in a speed range of 21–24 km/h and >24 km/h. In addition, all these variables were analysed for the 1st and 2nd half and the full match.

On the other hand, independent variables were considered those that describe the context in which players develop their competitive activity and that could determine their motor response: game position (central defenders (CD), external defenders (ED), central midfielders (CM), external midfielders/wingers (W), and forwards (F)), parts of the match, (first or second half), round number of league, and time of the season (pre-COVID and post-COVID).

### 2.4. Statistical Analysis

The data are described by arithmetic mean (M) and standard deviation (SD). The normal distribution of the variables was checked using the Kolmogorov–Smirnov and Shapiro–Wilk tests. As the variables were not distributed normally, the Wilcoxon test was used to compare pre-isolation and isolation periods by the number of played matches and by the playing position. When statistically significant differences were found, the size effect was estimated by using the Cohen’s d, Hedges’ G, Rank—Biserial Correlation and ∆ Glass index establishing two cut-off points: medium effect (0.30) and large effect (0.60) [4]. The confidence interval was set at 95% for the size effect. IBM SPSS Statistics software (Version 25.0. IBM Cor., Armonk, NY, USA) was used to make the mathematical calculations. The level of significance was set at *p* < 0.05.

## 3. Results

### 3.1. Differences between Pre-and Post-Isolation Periods by Number of Played Matches

The analysis revealed the following differences for those players who played the same number of matches before and after the isolation period. The minutes played were increased after the isolation period for players participating in one match (X = 2.23; *p* = 0.026), in two matches (X = 1.99; *p* = 0.047) and in three matches (X = 3.18; *p* = 0.001). Moreover, when the analysis was carried out for all players together weighted by the number of matches, there was an increase of the minutes played (X = 4.43; *p* < 0.001) and a reduction of the distance in the first half per minute (X = −2.72; *p* = 0.007), at HIR (X = −2.69; *p* = 0.007), at HIR in the second half (X = −2.26; *p* < 0.024), mean speed (X = −2.23; *p* = 0.026) and mean speed in the first half (X = −2.06; *p* = 0.039). The rest of the comparisons showed no differences *p* > 0.05, (see Table 2).

Additionally, for those footballers who played both before and after the isolation period, with a different number of matches between pre- and post-periods, the results showed a reduction in number of matches (X = −2.21; *p* = 0.027), DT (X = −2.11; *p* = 0.035), distance in the first half per minute (X = −4.08; *p* < 0.001), distance in the second half per minute (X = −2.06; *p* = 0.040), maximum speed (X = −2.38; *p* = 0.017) and maximum speed in the second half (X = −3.81; *p* < 0.001). The rest of the comparisons showed no differences at *p* > 0.05, (see Table 2).

### 3.2. Differences of Pre-and Post-Isolation Periods by Game Position

The analysis carried out by game position for all players together, balanced by the number of matches, showed an increase in the total minutes played after the isolation period by the central defenders (X = 3,02; *p* = 0.002), outside defenders (X = 2.52; *p* = 0.012) and midfielders (X = 1.96; *p* = 0.049). Additionally, the outside defenders incremented the distance run in the first half (X = 1.96; *p* = 0.049). On the contrary, the central defenders reduced the total distance run, covering between 14 and 21 km/h (X = −2.54; *p* = 0.011), and the second half distance run, covering between 14 and 21 km/h (X = −2.41; *p* = 0.016). The rest of the comparisons showed no differences *p* > 0.05, (Table 3).

Regarding the analysis by game position of all players together, not balanced by the number of matches they played before and after the isolation period, the results showed a general decrease after the isolation period in the following variables and game positions (see Table 4). (1) Central defender: distance first half/min (*p* = 0.039) and maximum speed second half (*p* = 0.031); (2) outside defender: second half distance >24 km/h (*p* = 0.015) and total sprint >24 km/h second half (*p* = 0.046); (3) midfielder: total distance/min (*p* = 0.035), distance first half/min (*p* = 0.006), total distance between 14–21 km/h (*p* = 0.025), total distance >21 km/h (*p* = 0.028), total distance >21 km/h/min (*p* = 0.010), distance >21 km/h second half (*p* = 0.031), distance >21 km/h/min 2nd half (*p* = 0.028), maximum speed (*p* = 0.007), maximum speed second half (*p* = 0.008), mean speed (*p* = 0.035) and mean speed 1 forst half (*p* = 0.016); (4) winger: the matches played (*p* = 0.007), total distance (*p* = 0.013), distance first half/min (*p* = 0.026), distance second half/min (*p* = 0.009) and total sprints > 24 km/h first half (*p* = 0.048). For the rest of the comparisons, no differences were found *p* > 0.05.

## 4. Discussion

This study analysed the physical performance differences in a week of three matches before and after the first COVID-19 lockdown and La Liga suspension. The main findings show that the intensity of pre-and post-COVID matches was not severely affected. This fact rejects the hypothesis that states that after a period of severe confinement during a pandemic there would be a strong detraining and a difficult adaptation to competitions.

Our results showed a slight increase in the total minutes of the footballers who played the same number of matches before and after the lockdown. This could be due to the number of interruptions that took place at the competition re-start caused by refreshment breaks during the games and the changes in the schedules that were made to reduce physical activity in the hours with higher atmosphere temperatures and to promote less physical load during the matches, [34] the rule change in the number of substitutions [35] or its influence in the game (related to the performance of the matches, due to ball possession, shots and ball retrievals) [36] and the consolidation of the Video Assistant Referee (VAR) in competitions [37].

However, the total distance travelled in the first half, the total distance between 14 and 21 km/h, the total distance in the second half between 14 and 21 km/h, the total number of sprints in the match and in the first part were reduced in the post lockdown period. This decline–decrease in performance coincides with the findings of Chmura et al. [38], who reported that the worst phases in terms of physical performance occurred after a long break and the last matches of the championship. Thus, during the traditional season, the detraining phase happens at the end of the league competition due either to an injury or to an illness. These training situations are common and are not comparable to the situation caused by the confinement due to the COVID-19 pandemic, despite the training conducted at home [19]. After the lockdown period, detraining effects of isolated training were expected, but the existing models do not fully describe the unprecedented conditions imposed by COVID-19 [39]. Inadequate periods of football-specific on-field training may be the cause of the increased incidence of injuries reported after the resumption. However, high-intensity training with a reduced amount of training after the competitive season may prevent a reduction in fitness [40,41]. In addition, due to the congestion of matches on the season’s schedule, coaches needed to rotate the players in the line-ups. Nevertheless, there was a significant number of players who accumulated more minutes and starting positions, diminishing their performance [42].

On the other hand, when the analysis was carried out without balancing the number of matches played in the pre-and post-isolation periods, there was a decrease in the number of matches played, total distance covered, distance in the first half per minute of play and total distance and maximum speed in the second half. Interestingly, the main differences were found when the data were analysed by game position. Specifically, the central midfielders were the players who suffered the most variations in terms of physical performance after the confinement. In this sense, previous literature has shown that midfielders are used to covering a significantly greater total distance compared to players in any other position and even run more total distance at high intensity than central defenders and forwards [15]. Furthermore, the external defenders and forwards showed greater sprint capacity and better agility, while the central defenders covered the shortest total distance and presented the shortest sprint time [42]. This fact could be associated with a better performance in intermittent exercise tests [43] and higher levels of maximum oxygen consumption by the defenders and midfielders [42].

In addition, the midfielders were the players most frequently present in the starting line-up. However, due to the type of efforts made in this game position on the field [15], the midfielders were more likely to decrease their physical performance values during the return to competition, as shown by our results, with medium to large size effects (see Table 4). Midfielders usually accumulate more playing minutes and probably are those who rotate less in the initial alignments [15]. On the contrary, the players in the last line of attack are the most substituted, probably because of the type of effort they make: more sprints and accelerations [36]. However, the new change in the rules produced by the pandemic [35] has allowed coaches to take advantage of the five substitutions to modify their line of attack to keep the goal scorers without fatigue and, consequently, avoid losses in their intensity actions [34].

Although our study has analysed a large amount of game intensity data from professional players of “La Liga Smartbank” during competition situations and there is limited literature analysing the competition restart after the COVID-19 stop in soccer, there are some limitations that should be mentioned. The pre-COVID-19 rounds choice was limited to rounds 8th, 9th and 10th because these were the only ones that were played during the same week (three matches in the same week) before the lockdown, while the post-COVID-19 rounds were the first ones of the league restart. Accordingly, this round selection could have had some influence on the physical state of the players. Lastly, these results could open new future investigation lines related to the causes of the maintenance of high-intensity action, which may be related to the new rule of five substitutions.

## 5. Conclusions

For all these reasons, we can conclude that the playing action’s intensity in a period after long-term confinement did not lead to a significant decrease in the performance indicators, despite the congestion of three matches per week. On the other hand, significant differences were found when the analysis was made by playing position. Specifically, the midfielders were the most used players, with the higher decrease in performance.

## 6. Theoretical and Practical Applications

As a theoretical implication, this study provides information about physical performance after a prolonged periods of inactivity caused by a so-called “black swan”, i.e., a surprise event with a great impact, which, once the event has passed, is rationalized by hindsight (making it seem predictable or explainable and giving the impression that it was expected to happen) [44]. An example of this could be new periods of confinement due to pandemics or wars, as in the recent case of the Russian–Ukrainian situation, where athletes have been suspended from any kind of sports activity [45].

As a practical application, coaches should take into account different solutions to maintain the intensity levels of their teams, such as choosing rotations in the starting line ups and making correct substitutions. This could be a key point to obtain the best conditional performance in the competitions. Efforts are becoming more and more intense [46], and it is therefore of interest to know what happened after COVID-19 to be able to make predictions. Therefore, this will affect the planning of training, where more attention should be paid to intermittency. In addition, the design of individualized and adequate training programs after long periods of confinement could be critical for players’ performance at the restart of the competitions.

## Figures and Tables

**Figure 1 ijerph-19-04252-f001:**
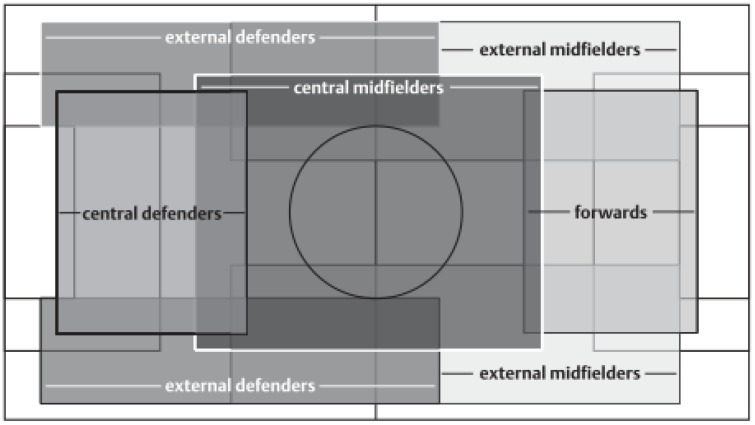
Tactical allocation of game positions based on match analysis 11.

**Table 1 ijerph-19-04252-t001:** Descriptive sample distribution.

	1 Match		2 Matches		3 Matches		All Matches Balanced		All Together	
	N	%	*n*	%	*n*	%	*n*	%	*n*	%
Central defender	4	16.7	6	60	8	57.1	18	37.5	61	21.3
External defender	4	16.7	2	20	2	14.3	8	16.7	61	21.3
Central midfielder	7	29.2	2	20	3	21.4	12	25.0	70	24.5
Winger	7	29.2					7	14.6	52	18.2
Forwards	2	8.3			1	7.1	3	6.3	42	14.7
Total players	24	100.0	10	100	14	100.0	48	100.0	286	100.0

Notes: *n* = number of participants; % = percentage; 1 match = players who played 1 match before and after the isolation period; 2 matches = players who played 2 matches before and after the isolation period; 3 matches = players who played 3 matches before and after the isolation period; all matches balanced = players who played the same number of matches (1, 2 or 3) before and after the isolation period; All together = players who played a different number of matches (1, 2 or 3) before and after the isolation period.

**Table 2 ijerph-19-04252-t002:** Comparisons of pre- and post-isolation periods by number of matches.

	Variable	PRE-COVID-19		POST-COVID-19		*p*		Effect Size		IC—95% *rbc*		IC—95% *Hedges’G*	
		*M*	*SD*	*M*	*SD*		*rbc*	*Hedges’ G*	*CLES*	*LL*	*UP*	*LL*	*UP*
One match	Total Minutes	96.79	2.08	98.48	2.16	0.026	−0.520	0.80	0.71	−0.775	−0.118	0.21	1.39
Two matches	Total Minutes	190.97	2.93	194.83	3.69	0.047	−0.709	1.16	0.79	−0.919	−0.184	0.21	2.11
Three matches	Total Minutes	289.09	3.35	294.88	3.01	0.001	−0.962	1.82	0.90	−0.988	−0.880	0.94	2.70
All together balanced by number of matches	Total Minutes	172.50	84.00	175.83	85.80	0.001	−0.735	0.04	0.51	−0.852	−0.547	−0.36	0.44
	Distance 1st half per min	117.52	33.00	106.65	9.93	0.007	0.451	−0.45	0.62	0.159	0.670	−0.85	−0.04
	Distance at 14–21 km/h (HIR)	2204.86	516.70	2079.88	592.03	0.007	0.446	−0.23	0.56	0.153	0.666	−0.63	0.18
	Distance at 14–21 km /h (HIR) 2nd half	1071.12	257.92	1011.03	321.05	0.024	0.374	−0.21	0.56	0.068	0.616	−0.61	0.20
	Total mean speed	8.75	0.58	8.62	0.63	0.026	0.369	−0.22	0.56	00.062	0.612	−0.62	0.19
	Mean speed 1st half	8.95	0.65	8.77	0.69	0.039	0.342	−0.27	0.58	0.031	0.592	−0.67	0.13
All together not balanced by number of matches	Matches played	1.26	1.08	1.06	1.05	0.027	0.145	−0.19	0.55	0.013	0.273	−0.35	−0.02
	Total distance (DT)	18,260.08	8405.43	17,223.80	7855.08	0.035	0.094	−0.13	0.54	−0.143	0.320	0.33	0.08
	Distance 1st half per min	118.82	33.49	105.73	12.05	<0.001	0.495	−0.51	0.64	0.296	0.653	−0.71	−0.30
	Distance 2nd half	9056.10	4210.00	8544.53	3953.65	0.040	0.250	−0.13	0.54	0.017	0.457	−0.33	0.08
	Maximun speed	30.62	1.70	30.04	2.94	0.017	0.289	−0.25	0.57	0.060	0.490	−0.45	−0.04
	Maximun speed 2nd half	29.73	1.82	28.85	2.98	<0.001	0.452	−0.36	0.60	0.256	0.628	−0.57	−0.16

Notes: *M* = mean; *SD* = standard deviation; *p* = level of significance; *Hedges’ G* = Effect size; *CLES* = Common Language Effect Size; rbc = Rank-Biserial Correlation; IC—95% = Interval confidence—95%; *LL* = lower limit; *UP* = upper limit. One match *n* = 24; Two matches n = 10; Three matches n = 14; All together balanced by number of matches n = 48; All together not balanced by number of matches (Matches played n = 286; rest of variables n = 200 PRE-COVID-19. n = 176 POST-COVID-19).

**Table 3 ijerph-19-04252-t003:** Analysis by play position for all players together, weighted bythe number of matches played before and after the isolation period.

	Variable	PRE-COVID-19		POST-COVID-19		*p*		Effect Size		IC—95% *rbc*		IC—95% *Hedges’G*	
		*M*	*SD*	*M*	*SD*		*rbc*	*LL*	*UP*	*LL*	*UP*	*LL*	*UP*
Central defender	Total minutes played	214.15	77.99	218.16	79.41	0.002	−0.813	−0.930	−0.543	−0.930	−0.543	−0.874	−0.975
	Total distance travelled between 14–21 km/h (HIR)	1861.10	392.86	1678.76	309.70	0.011	0.684	0.301	0.877	0.301	0.877	1.455	0.424
	2nd half distance travelled between 14–21 km/h (HIR)	881.95	194.69	775.96	143.93	0.016	0.649	0.242	0.862	0.242	0.862	1.565	0.327
Outside defender	Total minutes played	167.74	85.89	173.25	86.71	0.012	−1.000	−1.000	−1.000	−1.000	−1.000	−1.322	−1.45
	Distance run in the 1st half	8592.91	4134.55	8840.72	4146.94	0.049	−0.778	−949	−0.256	−949	−0.256	−1.326	−1.446
Midfielders	Total minutes played	160.26	85.27	162.90	86.84	0.049	−0.641	−885	−0.119	−885	−0.119	−1.101	−1.162

Notes: *M* = Note: *M* = mean; *SD* = standard deviation; *p* = level of significance; *rbc* = Rank Biserial Correlation; IC—95% = Interval confidence —95%; *LL* = lower limit; *UP* = upper limit. Central defender *n* = 18; Outside defender *n* = 8; Midfielders *n* = 12.

**Table 4 ijerph-19-04252-t004:** Analysis by play position of all players together, not weighted by the number of matches before and after the isolation period.

Play Position	Variable	PRE-COVID-19		POST-COVID-19		*X*	*p*	Effect Size			IC—95%*rbc*		IC—95% *Hedges´G*	
		*M*	*SD*	*M*	*SD*			*rbc*	*d*	∆ Glass	*LL*	*UP*	*LL*	*UP*
Central defender	Distance 1st half per min	103.01	7.60	100.28	5.46	−2.06	0.039	0.502	−0.41	−0.50	0.075	0.774	1.26	0.43
	Maximum speed 2nd half	29.35	1.50	28.40	1.75	−2.15	0.031	0.526	−0.58	−0.54	0.107	0.786	1.44	0.27
Outside defender	2nd half distance >24 km/h (HSR)	166.11	63.63	142.10	69.23	−2.44	0.015	0.527	−0.36	−0.35	0.161	0.766	1.20	0.48
	Total sprint >24 km/h (HSR) 2nd half	8.39	2.94	7.36	3.55	−2.00	0.046	0.447	−0.32	−0.29	0.042	0.726	1.16	0.53
Midfielders	Total distance per min	112.68	7.75	107.83	12.43	−2.11	0.035	0.567	−0.47	−0.39	0.117	0.824	1.32	0.38
	Distance 1st half per min	126.30	33.24	111.38	13.21	−2.72	0.006	0.731	−0.59	−1.13	0.384	0.897	1.44	0.26
	Total distance between 14–21 km/h (HIR)	2683.65	535.75	2462.62	608.77	−2.24	0.025	0.602	−0.39	−0.36	0.169	0.841	1.23	0.46
	Total distance >21 km/h (HIR)	436.67	161.96	372.07	138.99	−2.20	0.028	0.591	−0.43	−0.47	0.151	0.835	1.27	0.42
	Total distance >21 km/h (HIR) per min	4.55	1.70	3.80	1.38	−2.59	0.010	0.696	−0.48	−0.54	0.321	0.882	1.33	0.36
	Distance >21 km/h (HIR) 2nd half	212.76	89.01	177.95	84.75	−2.16	0.031	0.579	−0.40	−0.41	0.134	0.830	1.25	0.44
	Distance >21 km/h (HIR) min 2nd half	4.30	1.85	3.58	1.61	−2.20	0.028	0.591	−0.42	−0.45	0.151	0.835	1.26	0.43
	Maximum speed	29.84	1.35	28.54	2.54	−2.68	0.007	0.719	−0.64	−0.51	0.363	0.892	1.50	0.22
	Maximum speed 2nd Half	28.96	1.19	27.36	2.78	−2.64	0.008	0.708	−0.75	−0.58	0.342	0.887	1.61	0.12
	Mean speed full match	9.03	0.53	8.70	0.86	−2.11	0.035	0.567	−0.46	−0.38	0.117	0.824	1.31	0.39
	Mean speed 1st half	9.26	0.52	8.90	0.88	−2.42	0.016	0.649	−0.50	−0.41	0.242	0.862	1.35	0.35
Winger	Matches played	1.21	1.04	0.67	0.68	−2.69	0.007	0.400	−0.62	−0.79	0.111	0.627	1.47	0.24
	Total distance (DT)	22,559.63	7317.7	13,667.21	4719.63	−2.48	0.013	0.752	−1.44	−1.88	0.366	0.918	2.38	−0.51
	Distance 1st half/min	10,595.34	4275.16	6855.42	2479.55	−2.23	0.026	0.410	−1.07	−1.51	−0.158	0.744	1.96	−0.18
	Distance 2st half/min	11,195.77	3666.11	6811.79	2254.68	−2.61	0.009	−0.295	−1.44	−1.94	−0.716	0.282	2.38	−0.50
	Total sprints >24 km/h 1st half	10.46	3.77	9.00	3.28	−1.98	0.048	0.600	−0.41	−0.45	0.098	0.859	1.26	0.43

Notes: *M* = mean; *SD* = standard deviation; *p* = level of significance; *d* = Cohens’ *d* effect size; ∆ Glass = Delta Glass effect size. *LL* = lower limit; *rbc* = Rank-Biserial Correlation; IC—95% = Interval confidence—95%; *UP* = upper limit. Central defender *n* = 22; Outside defender *n* = 28; Midfielders *n* = 18; Winger *n* = 14.

## Data Availability

Not applicable.

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
