# Peer review of "COVID-19 Confinement Effects on Game Actions during Competition Restart in Professional Soccer Players"

_ijerph, 2022, doi:10.3390/ijerph19074252_

Round 1

Reviewer 1 Report

Thank you for allowing me to review this manuscript. While the overall study looks empirically supported by the data analysis, I'm not sure the value of this study. This study explored soccer players' performance differences pre- and post-covid and found no significant difference. Since coaches and players did their best to maintain or improve their performance even in the pandemic, the results may not be surprising. Also, while the authors provided previous COVID research, the previous studies might speculate players' performance will be decreased because covid restricts competitions and players' training. I couldn't see relevant evidence of decreasing players' performance after the pandemic in this current study's literature review and hypothesis. I have provided my specific comments to help strengthen the quality of this manuscript. 

First of all, I am concerned about the theoretical implications of this study. While the authors mentioned previous literature in the introduction, I couldn't find an overarching theory in this study. Additionally, there is no literature review section, so I'm not sure this study is supported by robust theoretical background. Lastly, there are practical applications (or implications) after the discussion section. However, there is no theoretical implications section. I understand that some studies find it challenging to establish theoretical or practical implications per the topics. However, the author needs to clarify how this study theoretically contributes to the literature of the research area. It would reinforce the value of this study. 

My second primary concern was the coverage of literature for the hypothesis. While the authors provided the gap of research from lines 55 to 67, the literature [8,13,15,16,17,18,19] was not related to football players' performance differences in pre- & post-covid. I can see the authors cited previous studies [1~5] regarding COVID-19 in football; those were used to who the difficulty of training regularly and postponement of sports competitions. I couldn't see the relevant support for "a period of severe confinement during the pandemic there will be a strong detraining and a difficult adaptation to the competition. (Line 222-223)." The authors' expectation before the findings might be opposite to the hypothesis. The authors need to clearly present the hypothesis or research question with relevant support of a proper literature review. 

As minor comments, "Soccer coaches...(line 25)" The authors used soccer and football together in this manuscript. In North America, the term football refers to American football, which is a different sport from soccer. I know that football and soccer are generally used in Europe and Asia (and even South America). Although this is a cultural difference, please consider using soccer throughout the manuscript. This is because European or Asian readers will not confuse with soccer, but North American readers may confuse with football. 

(Line 38) "isolation, witht opposition..." typo. 

Reviewer 2 Report

General Comments
This is an interesting and timely article.  A careful review f the English might be appropriate.  .  

Minor Comments:
    Page    Line                Comments
    1    38    Should that be “without” instead of “witht”?
    2    44    “this data” should be “these data” and “depends” should be “depend”.
    2    49    “ran” should be “run” and would t be better to say “shorter” rather than “smaller”?
    2    51    “…high intensity runs [11], 847 ± 349 m…”
    2    52    Should it be footballers’ since it is possessive?
    2    52-54    This sentence is a bit unclear.  What does it mean to be “associated to reach maximum speeds”?  And how is seed associated with the distance travelled?
    2    55    Does this mean the distance run and the sprint speed are the variables with the highest coefficients? 
    2    68    Should this read “…..previously mentioned, it seems advisable to compare ….?  Then you would not need the ending phrase “…is something new and interesting.”

    2    79    “The purpose of this study was ….”
    2    71    What does it mean “..characteristics of the actions….?  To what do “actions” refer?
    2    80-81    Does this mean the classifications were carried out following those recommended by Di Salvo et al. [1]?
    3    89    “…goalkeepers were excluded since their physical demands …..”
    3    103    “LiLigTM authrorized the use….”
    3    105-106    The Mediacoach physical performance data records the position….”
    3    112    “….allowing the intensity of the athlete’s run to be….”
    6    200    Does this mean “…all players together not equated for the number of matches played….”?
    8    228    There seems to be no need to enclose the phrase in () since it is an important fact.
    8    230    “….matches [27], the rue change….”
    8    238    “…..with the findings of Chmura et al. [31] who reported….”
    9    252    “…when the data were analyzed….”
    9    269    Should that be “…more sprints and accelerations…”?
    9    270    “…by the pandemic [28] have allowed….”
    9    275    Would “large” sound better than “big”, as in “…has analyzed a large amount of game….”?
9    279    Would it sound better to say:  “…because these were the only ones that were played….”?
9    294    Would this sound better to say “…levels of their teams, such as choosing rotations in the starting line ups and making correct….”?

Reviewer 3 Report

The manuscript is interesting for the topic. The sample is vast, only the design seems reductive for too solid conclusions.

14: This study analyzed 551 players…

23: Results seemed to show.. Conclusions too decisive for study design

25: “the most involved”

25: “The main findings of this study could provide insight to football coaches for rotations in starting lineups and in-game substitutions, to not affect the intensity levels of competition”

66: There are actually studies in the literature that have evaluated the impact of COVID-19 lockdown and schedule congestion, this suggested statement and ref can serve as a rationale for your manuscript (maybe discussion)...

“while the overall distance run and that covered at lower intensities varied across games, high-intensity running performance and injury risk were generally unaffected during a prolonged period of fixture congestion (ref https://doi.org/10.1055/s-0031 -1283190 ). Furthermore, even when the congestion fixture created by the first COVID-19-related lockdown was analyzed, there was no significant impact on muscle injury rates in professional players (ref: http://dx.doi.org/10.23736/S0022-4707.21 .12903-2 ). Although probably in the long term some authors have detected more injuries during the 2020-2021 season affected by COVID than in the previous two seasons, perhaps demonstrating a link between fixture congestion and athlete injuries (ref: https://doi.org/10.1080 /00913847.2021.1980746 ). Therefore, the stop due to the lockdown for COVID-19 did not provide evidence on the decrease in physical performance in the post-lockdown period compared to the pre-lockdown period, nor if it is caused by the accumulation of games (congested time).ref : https://doi.org/10.3390/ijerph18073685 .

93 move Descriptive sample distribution in results section. In methods section describe each methodology

102-110 There are previous works that have used these software

158 unfortunately for non-parameter tests the rank biserial correlation must be used as effect size

Table 2 not captivating .. represent the mean as mean ±SD, If you have used the Wilcoxon why is there a Z?

220 first COVID-19 lockdown and La-Liga suspension, not just “confinement”

287 does not appear to have led to a significant decrease in performance. Modify confinement, especially if a separate noun because when it is not linked to the pandemic … it is inappropriate

Round 2

Reviewer 1 Report

The authors addressed all of my previous suggestions well. I was concerned about the theoretical background, and the authors revised it with additional related references, such as from 18 to 26. Especially, the citations of García-Aliaga et al. (2021) and Mannino et al. (2021) were what I was looking for the relevant evidence to support decreasing players' performance after the pandemic in my first review. While I didn't comment, the methods and results sections have been improved by the authors' revisions. 

Also, I appreciate the authors added the section of "Theoretical and practical Applications." Not only [45] and [46], it would be better if the authors state [25] and [26] with extending the studies regarding the impact of COVID-19 in sports leagues. 

Overall, the manuscript has been improved, but some typos should be revised. For example, in line 308, "Theoretical and practical Applications," "p" should be the upper case, or "A" should be the lower case. So, lastly, I suggest professional proofreading and revision of another reviewer (s). I hope the best of luck with the authors' research. 

Author Response

We would like particularly thank the reviewer for the time spent reviewing the paper as well as for the contributions and suggestions that have improved the quality of the work. In accordance with the suggestions provided, the typographical error has been corrected and has been corrected by a professional who has checked the errors that occurred with the language.

Reviewer 3 Report

The manuscript has improved significantly and I can suggest suitability to publication

Author Response

We would like particularly thank the reviewer for the time spent reviewing the paper as well as for the contributions and suggestions that have improved the quality of the work. Thank you for recommending it for publication